# Comparative Study of the Influence of Bio-Resin Color on the Dimension, Flatness and Straightness of the Part in the 3D Printing Process

**DOI:** 10.3390/polym13091412

**Published:** 2021-04-27

**Authors:** Aurel Tulcan, Mircea Dorin Vasilescu, Liliana Tulcan

**Affiliations:** 1Department of IMF, Politehnica University Timisoara, 300006 Timisoara, Romania; aurel.tulcan@upt.ro; 2Department of MMUT, Politehnica University Timisoara, 300006 Timisoara, Romania; mircea.vasilescu@upt.ro

**Keywords:** DLP printing, 3D measurement, surface deviations, flatness, straightness, analysis of variance, response surface design

## Abstract

This paper aims to determine whether the color of based-plant resin called, by the manufacturer, eco-resin has an influence on the dimensions and geometric accuracy of the 3D-printed part. The analysis of flatness, straightness and dimensions deviations was carried out with high-precision measurement systems, and according to current standards regarding linear dimensions and geometrical tolerances. A coordinate measuring machine with contact probes was used to measure the printed part’s physical characteristics, and analysis of variance and response surface design methods were used for the data analysis. The printing experiment was carried out for each color. After that, the measurement of the printed parts and the study of the data were performed. The first finding is that for black and clear eco-resin, there are problems with the printing of the supports. Based on standard data for the range of nominal lengths of the part for linear dimensions, flatness and straightness, the measurement results can be included in different tolerance classes within standard value limits. The best value of the printed structure was obtained for clear eco-resin. The paper demonstrates that the impact of the color of the eco-resin is more important than the supports density for all the studied features. Based on 3D measurements, the optimal values for each of the eco-resin colors regarding the flatness, straightness and linear dimensions deviations of the 3D printed part were also determined.

## 1. Introduction

Different processes can be used to make 3D printing parts. The digital light process (DLP) is considered in this study. This type of printing is important in relation to the printed surface quality and printing accuracy for different mechanical parts. This study continues the previous author’s research in [1] regarding the investigation into aspects of the technological features, which may influence the 3D printing process with resin material. It should be noted that this study was performed for standard resin, which can be compared with 3D printed parts with fuse deposit modelling (FDM) [2,3,4,5,6]. At the same time, it is important to note that the printing time is shorter for the digital light processing printing method, and 3D printed parts with different structures of support are possible [7,8,9,10].

This article studies whether it is an influence in the technological construction of the printed part. It is essential to determine this aspect concerning the 3D printed parts by the elements of its design. The main aspects analyzed were how dimensional elements and surface characteristics change if a part is 3D printed with a specific resin color.

The chemical composition and structure of eco-resin are different from the basic resin [11]. The most important aspect is that the eco-resin is more complex, considering the basic resin’s chemical composition. This is the reason why in this paper, the influence of this type of eco-resin and the dye in the part structure in relation to the technological characteristics of the 3D printed part has been studied.

A very accurate measurement method is required to determine the influence of the color of the eco-resin on the quality of the 3D printed part’s features. One of the most accurate measurement methods is the tactile measurement method [12,13]. The tactile method, though it is not so fast compared to the non-contact (optical) method, ensures the best accuracy of the measurement results in the micrometer range. Another reason for using the tactile measurement method is to avoid any influence of the color or the transparency of the ecological resin on the measurement results that could be influenced by using the optical method. The tactile measurement method has no required preparatory operations such as covering shiny, transparent or black parts [13]. In conclusion, the measurement of the physical characteristics of printed parts will be performed on a coordinate measuring machine (CMM) with tactile probes.

The main goal of the study consists of defining how the resin color influences the geometry of the printed part and the extent of deviations of size and shape occur. Based on the part’s size deviations, according to the ISO 2768-1 standard [14] regarding the tolerances for linear and angular dimensions without individual tolerance indications, the printed part will be included into a tolerance class, with a standard value limit, for the nominal lengths of the part over 30 mm up to 100 mm. In a similar way, the part’s geometrical shape deviations will be analyzed, and according to the ISO 2768-2 standard [15], regarding the geometrical tolerances for features without individual tolerance indications, the value of straightness and flatness deviations will be included in a tolerance class, with a standard value limit, for ranges of nominal lengths over 30 up to 100 mm. Finally, the present work allows to define the precision class that can be achieved by the 3D printing process using different eco-resin colors, regarding the dimensional and geometrical accuracy of parts manufactured with DLP technology.

## 2. Materials and Methods

### 2.1. Materials Used in the Experiment

#### Resin Used in the DLP 3D Printing Part

Five different color resin were used for the 3D printed resin. The color started from clear resin and finished with black. The authors want to determine if the resin’s color could generate modification regarding dimensional and geometrical parts accuracy. The resin used is in accordance with the 3D printer type and is an ANYCUBIC eco-resin [16].

From the point of view of the plant-based resin’s chemical structure [17], this is based on several specific components, mentioned below:


45% concentration of Fany acids, soya, epoxidized, Bu esters;30% isooctyl acrylate;15% 2-((2,2-Bis(((1-oxoallyl)oxy)methyl)butoxy)methyl)-2-ethyl-1,3-propanediyl diacrylate;5% 2-hydroxy-1-(4-(4-(2-hydroxy-2-methylpropionyl)benzyl)phenyl)-2-methylpropan-1-one;5% polychloro copper phthalocyanine.


From the resin’s composition, the vast majority of the composition is of plant type or vegetable resin (45%). It can also be observed that five percent of the composition is of the dye type. The last position is the green dye that is added to the basic structure of the resin. It can be noted that the existence of the suspended pigment may produce different effects on the behavior of the material during 3D printing with the resin of different colors.

### 2.2. 3D Printer, Post-Curing System and Measuring Machine Used in the Experiment

#### 2.2.1. 3D Printer and Post-Curing System Used for 3D Printing Part with Resin

The 3D printer used in this study was an ANYCUBIC PHOTON 3D printer with new software compared to the 3D printer used in the previous study [1]. This 3D printer can print a different part with a different setting for each element. The 3D printer was placed in a thermal chamber for good thermal and moisture stability for printing. For post-curing, a tank was used containing 70% sanitary alcohol, and in another tank, distilled water for the second step of curing the eco-resin. The first step of curing is made by immersing the printed part in the tank and, after this step, additional curing of a surface is made with a toothbrush on each part of the printed part. The polymerization of the surface was made by exposing the printed part to normal light for this study.

#### 2.2.2. Measuring Machine Used to Determine the Printed Part Geometry

For the measurement of the physical characteristics of the 3D printed parts, a DEA Global Advantage 7.10.7 bridge coordinate measuring machine (CMM) with tactile probes, shown in Figure 1a, was used [18]. The 3D printed part was clamped in a parallel vice with three pins [19] installed on the CMM table, shown in Figure 1b. As the CMM uses tactile probes, the different eco-resin colors used for the printed parts do not influence the accuracy of the measurements results. The maximum permissible error (MPEE) for length measurement, specified in the machine’s calibration certificate, is: MPEE = 1.9 + L/300 μm [18,20]. To create and execute the measurement routines to verify the geometry of the 3D printed parts, the PC-DMIS 2019 R2 metrology software was used [21].

### 2.3. Experimental Plan

#### 2.3.1. Determining the Time for Exposure at 3D Printing with Resin

This step is important for the printing process to achieve the characteristics of a good polymerization of the 3D printed part. This step is executed with a special program generated by ANYCUBIC software named RERF [22]. In this sense, from our point of view, the setting is 4 s for the first printed part, and the iteration was by 1 s for each step.

#### 2.3.2. Generating the Structure of the 3D Printing Part

This step takes into consideration the data obtained in the previous study [1]. It was determined that for computation, the optimum situation corresponding at 55% supports density with a contact depth of 0.25 mm and a diameter of the contact surface with 1.25 mm. The two extreme values taken into consideration were 50% and 60% for the density of supports, 1.2 mm and 1.6 mm for the contact surface diameter and 0.2 mm and 0.3 mm for contact depth. These values correspond to a zero number of broken supports. The data of the generation of the printed part are presented in Table 1. This table simultaneously presents the values that are important for the setting of the supports construction. The height of the layer was 0.05 mm for all colors.

It is possible to observe that the layer’s number is different from the point of view of layer generation. This aspect is important in the 3D dimension measuring of the part.

#### 2.3.3. Printing the Parts

For each color of the eco-resin, two parts were simultaneously printed in the first step, shown in Figure 2a, at the lower and upper value. The median value was printed in the second step, shown in Figure 2b, in the same conditions as the first two printed parts.

It is essential to note that the support structure was corrected to distribute these elements in this step. The correction was made at three levels. At the first level, the linear density of supports in the part’s sidewall was fixed. At the second level, the density of supports in the circular holes generated in the part was corrected. The last correction refers to the error generated by a lousy density of supports on the part’s flat surface. The software in which the supports were developed was ANYCUBIC PHOTON slicer 64 [23]. After this step, the structure was saved as a stereolithography file (STL). This structure was imported in the Photon Workshop V2.1.21 [24], generating the printing process layers.

#### 2.3.4. Study of the Dimensions and Geometry of the 3D Printed Part

The first step in creating the measurement routine was to import the part CAD model and perform the manual alignment of the printed part and then the automatic alignment. As the printed part has large deviations from the nominal geometry, automatic alignment accuracy is essential. The second step was to measure the analyzed features. Figure 3a shows the printed part’s measurement features, and Table 2 is defined as the printed part probing strategy. For each physical surface, the measurement feature (e.g., plane, straight line), Z-coordinate of the measuring plane and the number of the probing (hits) points is defined.

The probing point’s pattern for measuring flat surface and different contours (straight lines) and the deviation in each probing point are shown in Figure 3b. The arrows indicate the directions of the 3D-printed part deviations. The length of the arrows is proportional to the size of the deviations.

## 3. Results

### 3.1. Determining the Printing Condition and Printed Part Dimensions

#### 3.1.1. Determining the Exposure Time to 3D Printing with Resin

This step is essential to the printing process by the characteristics of printing. It is necessary to do this from time to time because it is possible to modify the printing process’s eco-resin polymerization characteristics. The new version of the ANYCUBIC 3D printer has a process dedicated to this aspect. The correct position can be determined by optical inspection of the same characteristics of the printed part. One of these characteristics is the testing text of the 3D printer, which is very fine and difficult to be printed.

The correct value for 3D printing was determined at which the platform lift speed on *Z*-axis was 2 mm/s. The platform retraction speed on the *Z*-axis for the non-black or unclear eco-resin was 5 mm/s; eight bottom layers were generated with 60 s bottom expose time, shown in Figure 4a.

In order to determine the optimal setting with the help of this software, several checks are made. The first is related to the integrity of the generated structure and the correct arrangement of the generated elements. It can be observed that for short exposure time, regardless of color, there are problems of geometrically correct generation. The second aspect is related to the elasticity or stiffness of the V elements. The third aspect is related to the written text under the V element, on the right side, by the software producer.

Based on these observations, the minimum exposure time was determined for each color, which is detailed as follows:


For green eco-resin, the value for which the 3D printing can give good printing results is between 8 and 9 s, determined for positions 5 and 6 from Figure 4b.For blue eco-resin, the value for which the 3D printing can give good printing results is 9 s, determined for position 6 from Figure 4b;For violet eco-resin, the value for which the 3D printing can give good printing results is 9 s, determined for position 6 from Figure 4b;For black eco-resin, the value for which the 3D printing can give good printing results is 10 s, determined for position 7 from Figure 4b. This value is determined by the supports section zone, in which a value of 8 s is not sufficient for the mechanical and polymerization strength. Additionally, the platform retraction speed on the *Z*-axis was reduced from 5 to 3 mm/s for the same mechanical consideration;For clear eco-resin, the value for which the 3D printing can give good printing results is 8 s, determined for position 5 from Figure 4b. An important aspect is the printing in parallel of two or multiple parts with different heights for the supports and other geometric structures. Figure 5 shows the heavy structure and the medium structure printed with 8 s polymerization time. In terms of the support’s integrity, the heavy structure provides good results, while the medium structure is affected. This observation is good for black and clear eco-resin.In Figure 6, it is possible to observe the structure after reprinting with 10 s for the polymerization time.


#### 3.1.2. Results Concerning the Body Dimensions of the Printed Part

In this experimental process, the printed part body’s side surfaces with five eco-resin colors were measured. Each side surface of the part was measured as a straight line, shown in Figure 7, in 14 probing points for length measurement and 12 probing points for width measurement, according to Table 2.

The input data for generating the printed part and the body size measurement results are presented in Table 3. The eco-resin color and the supports density were the experimental factors, and the length and width deviation were the response variables. The data is structured identically concerning the printing step.

The experimental data were analyzed with Minitab Statistical Software 17 [25]. The eco-resin color experimental factor was used as a color code from 1 up to 5, shown in Table 3. An analysis of variance (ANOVA) [25,26] was performed for both response variables. Table 4 presents the analysis of variance for length deviation.

If the *p*-value for experimental factors is less than 0.05, for a 95% confidence level, the experimental factors significantly affect the outcome of the response variable [26]. It can be seen that the length deviation is significantly affected by the color of the eco-resin, while the supports density does not have a significant influence on the length deviation.

Figure 8 shows the estimated response surface for length deviation, and Figure 9 shows the contours of the estimated response surface for length deviation. It can be seen that for green, blue and violet eco-resin, the length deviations are negative, and for black and clear eco-resin, the deviations are positive. The violet eco-resin with 55% and 60% supports density gives the minimum length deviation (−0.006 mm and −0.008 mm), which is very close to zero. This combination can lead to obtaining an actual length very close to the nominal length.

The regression of the fitted model for the printed part length deviation is presented in Equation (1):Length Deviation = 0.0024 + 0.1159 ResinColor_Black − 0.0074 ResinColor_Blue + 0.0603 ResinColor_Clear − 0.1541 ResinColor_Green − 0.0147 ResinColor_Violet − 0.0156 Density_50 + 0.0110 Density_55 + 0.0046 Density_60(1)

Table 5 presents the analysis of variance for width deviation. It can be seen that the width deviation is significantly affected only by the color of the eco-resin (*p*-value less than 0.05). At the same time, the supports density does not have a significant influence on the length deviation (*p*-value greater than 0.05).

The regression of the fitted model for the printed part width deviation is presented in Equation (2):Width Deviation = 0.0054 + 0.0926 ResinColor_Black − 0.0104 ResinColor_Blue + 0.0423 ResinColor_Clear − 0.1147 ResinColor_Green − 0.0097 ResinColor_Violet- 0.0054 Density_50 + 0.0122 Density_55 − 0.0068 Density_60(2)

Figure 10 shows the estimated response surface for width deviation, and Figure 11 shows the contours of the estimated response surface for width deviation. It can be seen that for green, blue and violet eco-resin, for almost all the support density combinations, the width deviations are negative. For black and clear eco-resin, the deviations are positive. The violet eco-resin with 55% supports density gives the minimum width deviation (0.003 mm), which is very close to zero. A good result regarding the width deviation (−0.005 mm) also provides blue eco-resin with 60% supports density. These combinations of eco-resin color and supports density can obtain an actual width very close to the nominal width.

#### 3.1.3. Results Concerning the Body Flatness of the Printed Part

In the experimental process carried out in this phase, the flat surface A, shown in Figure 12, was measured at 110 contact points, according to Table 2. The probing points were spread over the entire surface of the printed part.

The input data for the printed part’s generation and the measurement results regarding the surface flatness are presented in Table 6. The data is structured identically concerning the printing step.

The analysis of variance for flatness is presented in Table 7. As the *p*-value is less than 0.05 [26] for both experimental factors—resin color and supports density—they have a 95% statistically significant influence on the body surface’s flatness. The impact of eco-resin color on flatness is more important than that of the supports density.

The regression of the fitted model for the flatness of the body surface is presented in Equation (3):Flatness = 0.9839 − 0.2466 ResinColor_Black + 0.0674 ResinColor_Blue− 0.0883 ResinColor_Clear + 0.1684 ResinColor_Green + 0.0991 ResinColor_Violet + 0.0859 Density_50 − 0.0023 Density_55 − 0.0835 Density_60(3)

Figure 13 shows the estimated response surface for flatness, and Figure 14 shows the contours of the estimated response surface for the flatness of the printed part body sur-face. It can be seen that the green eco-resin gives the maximum value for flatness deviation (1.240 mm) with 55% supports density. The black eco-resin gives the minimum value for flatness deviation (0.722 mm) with 60% supports density. A good result (0.776 mm) also gives clear eco-resin 55% supports density.

#### 3.1.4. Results Concerning the Straightness of the Body Surface of a Printed Part

Regarding the straightness of the body surface of the printed part studied in this experiment, four linear contours noted with X1, X2 and Y1, Y2, shown in Figure 15, were measured along the *X*- and *Y*-axis. The linear contours were defined with an offset of 0.8 mm from the external edges of surface A inside the surface. The probing points for each contour are presented in Table 2.

The output data for the straightness of the printed part’s body surface, measured parallel to the *X*- and *Y*-axis, is presented in Table 8. The data is structured identically in relation to the printing step. The straightness analyze starts with the contour X1.

According to the analysis of variance for contour X1 straightness, presented in Table 9, both experimental factors—resin color and supports density—have a 95% statistically significant influence on the straightness of the X1 contour. For both experimental factors, the *p*-value is less than 0.05.

The regression of the fitted model for the contour X1 straightness is presented in Equation (4):Straightness_X1 = 0.25813 − 0.05447 ResinColor_Black + 0.02353 ResinColor_Blue − 0.06447 ResinColor_Clear + 0.07087 ResinColor_Green + 0.02453 ResinColor_Violet + 0.02487 Density_50 − 0.00493 Density_55 − 0.01993 Density_60(4)

Figure 16 shows the estimated response surface for the contour X1 straightness. Figure 17 shows the contour of the estimated response surface for the contour X1 straightness of the printed part body surface. It can be seen that the maximum value for straightness deviation is 0.357 mm and is obtained for green eco-resin with 50% supports density. The minimum value for straightness deviation is 0.174 mm. This is achieved for clear eco-resin with 60% support density. Good results are also obtained for black eco-resin: 0.193 mm straightness deviation for 55% support density and 0.195 straightness deviation for 60% supports density.

##### Contour X2

Table 10 presents the analysis of variance for contour X2 straightness. It can be seen that the contour X2 straightness is significantly affected only by the color of the eco-resin (*p*-value less than 0.05). At the same time, the supports density does not have a significant influence on the contour X2 straightness (*p*-value greater than 0.05).

The regression of the fitted model for the contour X2 straightness is presented in Equation (5):Straightness_X2 = 0.27487 − 0.0505 ResinColor_Black + 0.0141 ResinColor_Blue- 0.0689 ResinColor_Clear + 0.0805 ResinColor_Green+ 0.0248 ResinColor_Violet − 0.01667 Density_50 + 0.01613 Density_55+ 0.00053 Density_60(5)

Figure 18 shows the estimated response surface for the contour X2 straightness. Figure 19 shows the contour of the estimated response surface for the contour X2 straightness of the printed part body surface. It can be seen that the maximum value for straightness deviation is 0.378 mm, which is obtained for the green eco-resin with 60% supports density. The minimum value for straightness deviation is 0.186 mm, obtained for the clear eco-resin with 60% support density. A good result is also accepted for the clear eco-resin, 0.193 mm straightness deviation for 50% support density.

##### Contour Y1

According to the analysis of variance for contour Y1 straightness, presented in Table 11, both experimental factors—resin color and supports density—have a 95% statistically significant influence on the straightness of the Y1 contour. For both experimental factors, the *p*-value is less than 0.05. The impact of eco-resin color on contour Y1 straightness is more significant than that of the supports density.

The regression of the fitted model for the contour Y1 straightness is presented in Equation (6):Straightness_Y1 = 0.7145 − 0.1408 ResinColor_Black + 0.0759 ResinColor_Blue− 0.1811 ResinColor_Clear + 0.1875 ResinColor_Green + 0.0585 ResinColor_Violet + 0.0255 Density_50 + 0.0327 Density_55 − 0.0583 Density_60(6)

Figure 20 shows the estimated response surface for the contour Y1 straightness, and Figure 21 shows the contour of the estimated response surface for the contour Y1 straightness of the printed part body surface. It can be seen that the maximum value for straightness deviation is 0.971 mm, which is obtained for green eco-resin with 50% supports density. The minimum value for straightness deviation is 0.464 mm, obtained for clear eco-resin with 60% support density.

##### Contour Y2

Table 12 presents the analysis of variance for contour Y2 straightness. It can be seen that the contour Y2 straightness is significantly affected only by the color of the eco-resin (*p*-value less than 0.05). At the same time, the supports density does not have a significant influence on the contour Y2 straightness (*p*-value greater than 0.05).

The regression of the fitted model for the contour Y2 straightness is presented in Equation (7):Straightness_Y2 = 0.6475 − 0.1055 ResinColor_Black + 0.0971 ResinColor_Blue − 0.1702 ResinColor_Clear + 0.0831 ResinColor_Green+ 0.0955 ResinColor_Violet + 0.0083 Density_50 + 0.0479 Density_55 − 0.0561 Density_60(7)

Figure 22 shows the estimated response surface for the contour Y2 straightness. Figure 23 shows the contour of the estimated response surface for the contour Y2 straightness of the printed part body surface. It can be seen that the maximum value for straightness deviation is 0.862 mm, which is obtained for green eco-resin with 55% supports density. The minimum value for straightness deviation is 0.443 mm obtained for clear eco-resin with 60% support density.

## 4. Discussion

The first conclusion is to consider the correct exposure time for printing. It can be observed that the exposure time for a single printed part is consistent with all the eco-resin colors in the program to determine the exposure time. For multiple printed parts with a different set of supports, it is recommended that a high value be considered for the exposure time in the zone of supports or all printed part. This aspect is observed only for black and clear resin.

The second conclusion refers to the printed part body dimensions according to the ISO 2768-1 standard [14] regarding the linear and angular dimensions without individual tolerances indications. Figure 24 shows the overlaid contours plot of length and width deviation. In Figure 24a, the white area represents the optimal combination of the eco-resin color and supports density that gives negative deviations between 0 and −0.15 mm of printed part length and width. In Figure 24b, the white area represents the optimal combination of the eco-resin color and supports density that gives positive deviations between 0 and 0.15 mm of printed part length and width. For blue and violet eco-resin, the length and width deviations are small, the printed part dimensions being very close to the part nominal dimensions.

Based on these observations, it can be concluded that for all other combinations except green eco-resin with 50% supports density, the length and width of the printed part could be included in the f tolerance class, with a standard value of ± 0.15 mm, for the nominal lengths of the part over 30 mm up to 100 mm [14].

The third conclusion refers to the printed part body contours straightness according to the ISO 2768-2 standard [15], regarding the geometrical tolerances for features without individual tolerances indications. Figure 25 shows the printed part’s overlaid contours plot of contours straightness X1, X2 and Y1, Y2. The white area represents the optimal combination of the eco-resin color and supports density that gives straightness deviations up to 0.2 mm for X1 and X2 contours and up to 0.5 mm for Y1 and Y2 contours. These straightness deviation values could be obtained only for the printed part from clear eco-resin with 60% supports density.

Based on these observations, it is possible to conclude that the X1 and X2 contour straightness deviations of the printed part could be included in the K tolerance class, with a standard value of 0.2 mm. For the part’s nominal lengths, over 30 mm up to 100 mm, the Y1 and Y2 contour straightness deviations of the printed part are very close to the L tolerance class and could be included in this tolerance class with a standard value of 0.4 mm [15].

The fourth conclusion involves determining optimal conditions that will produce the best value for the printed part body shape deviations. Since flatness and straightness of the printed part flat surface are important in determining the printed part’s quality, these properties will be considered simultaneously. Figure 26 shows an optimal solution for the input variables.

The target value settings for input variables flatness and straightness are 0. The upper-value settings are 0.8 mm for flatness deviations, 0.2 mm for straightness deviations of the contours X1 and X2, and 0.5 mm for straightness deviations of the contours Y1 and Y2. In conclusion, the desirability can be obtained only for the combination of 60% supports density and clear eco-resin.

## Figures and Tables

**Figure 1 polymers-13-01412-f001:**
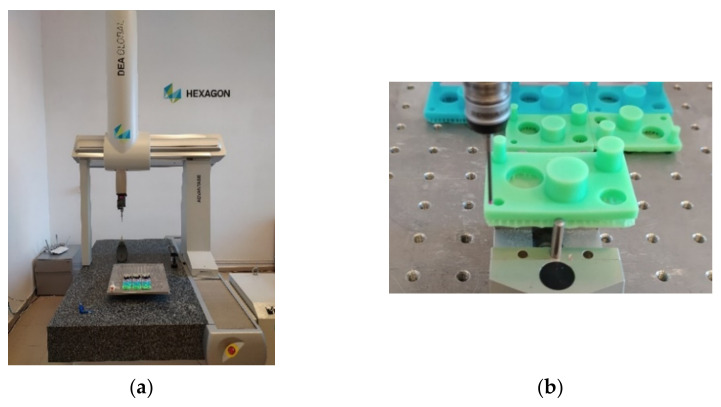
The coordinate measuring machine used to measure the printed parts: (**a**) view of the 3D printed parts; (**b**) clamping the part.

**Figure 2 polymers-13-01412-f002:**
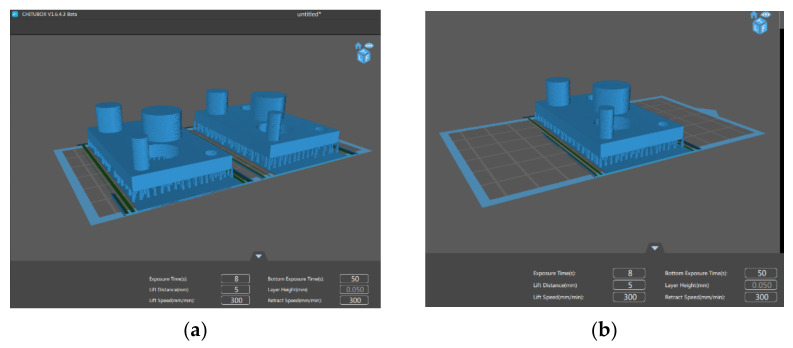
The generation of the 3D printed parts: (**a**) at lower and upper value; (**b**) at the median value.

**Figure 3 polymers-13-01412-f003:**
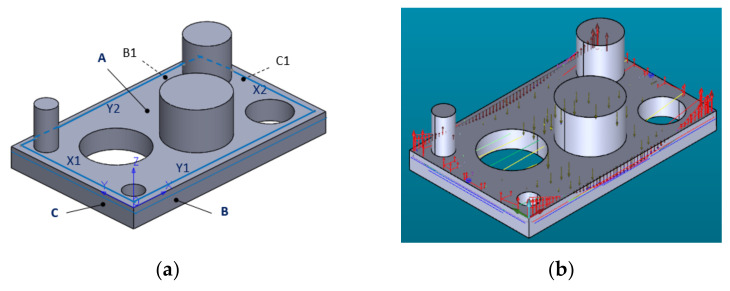
Printed part measurement: (**a**) Definition of the part measurement features: A, B, C, B1, C1—flat surfaces; X1, X2, Y1, Y2—linear contours; (**b**) Probing point’s patterns and deviations direction.

**Figure 4 polymers-13-01412-f004:**
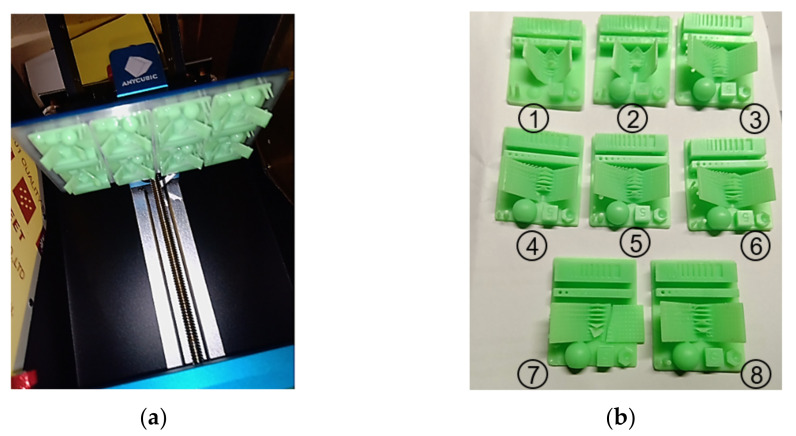
ANYCUBIC 3D printer and exposure time test: (**a**) Platform on *Z*-axis top position with eight simultaneously printed parts; (**b**) Positions on the printed parts platform.

**Figure 5 polymers-13-01412-f005:**
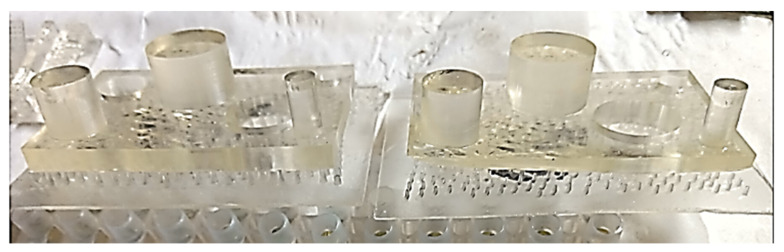
The clear parts printed with 8 s time: heavy structure on the left and medium structure on the right.

**Figure 6 polymers-13-01412-f006:**
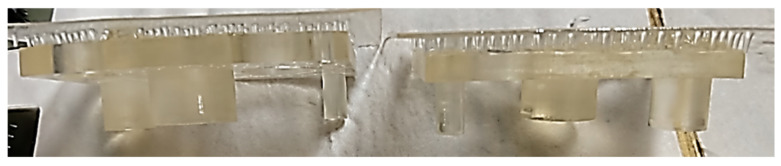
The clear parts printed with 10 s for heavy structure in the left position and 8 s for medium structure in the right position.

**Figure 7 polymers-13-01412-f007:**
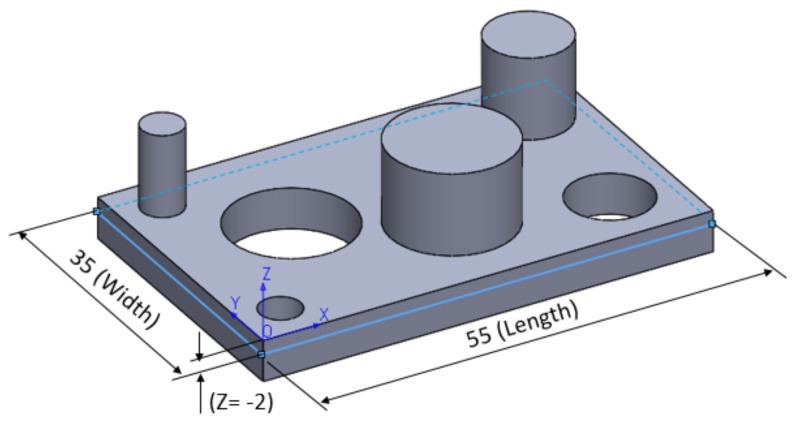
Definition of the length and width measurement.

**Figure 8 polymers-13-01412-f008:**
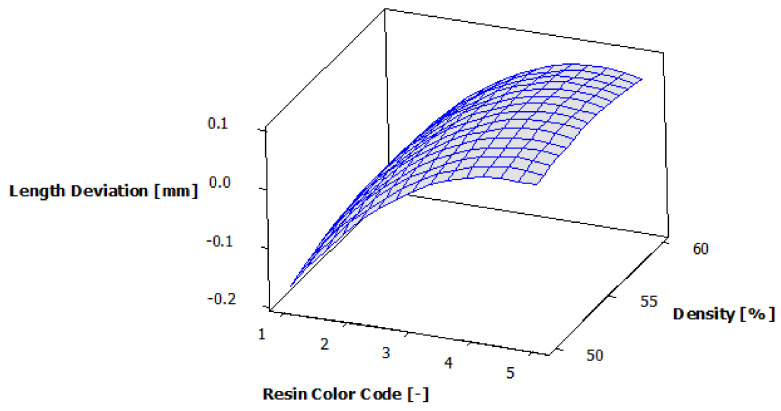
Estimated response surface for length deviation.

**Figure 9 polymers-13-01412-f009:**
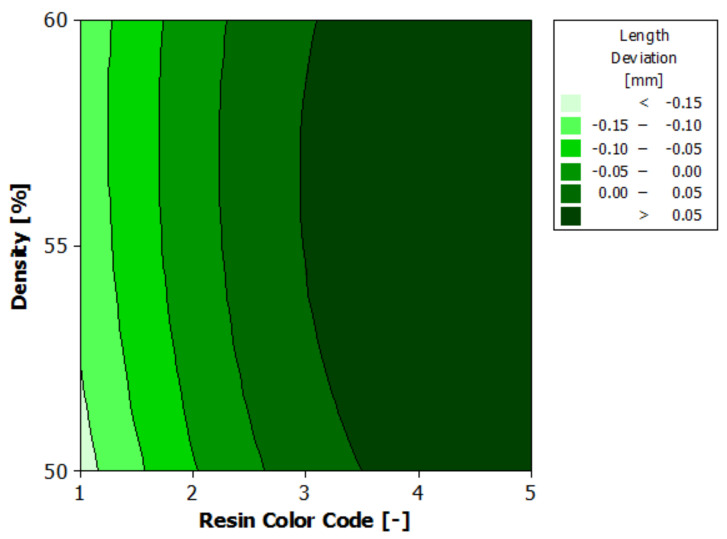
Contours of estimated response surface for length deviation.

**Figure 10 polymers-13-01412-f010:**
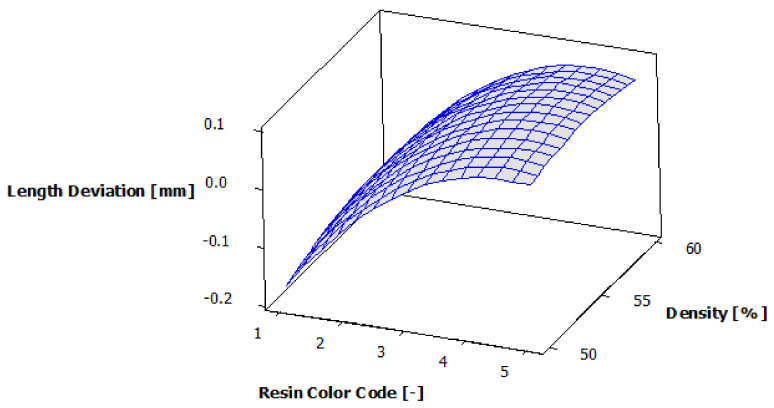
Estimated response surface for width deviation.

**Figure 11 polymers-13-01412-f011:**
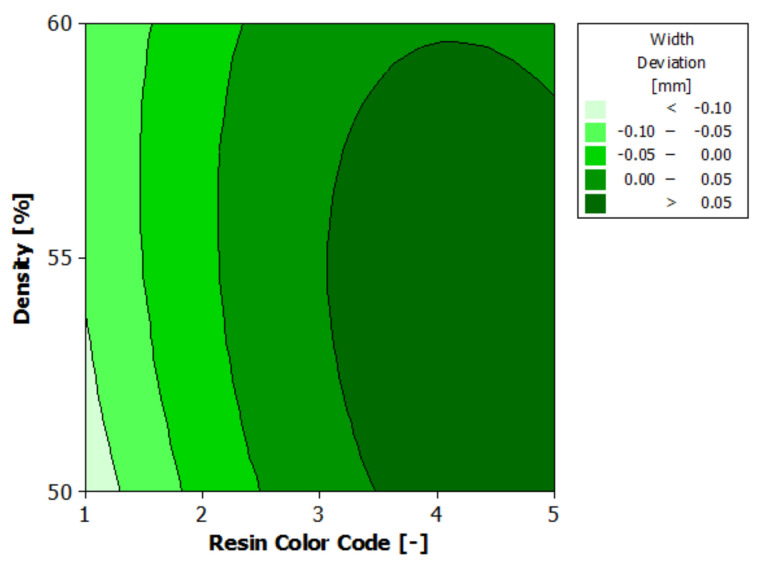
Contours of estimated response surface for width deviation.

**Figure 12 polymers-13-01412-f012:**
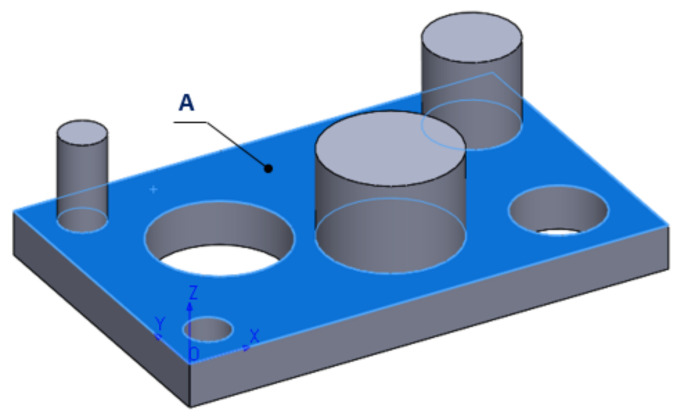
Definition of the measured flat surface.

**Figure 13 polymers-13-01412-f013:**
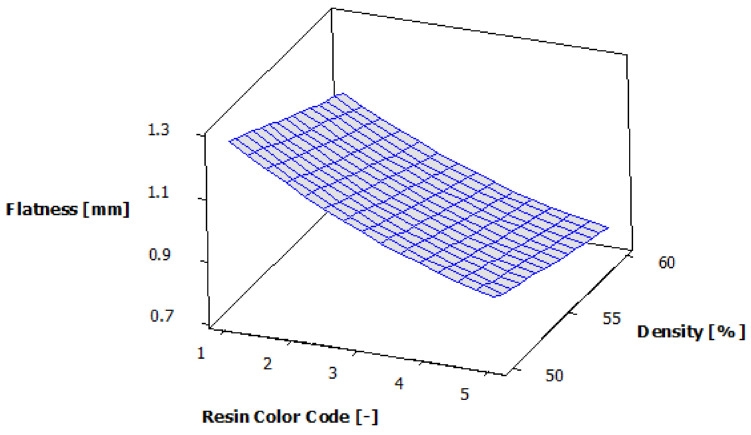
Estimated response surface for flatness.

**Figure 14 polymers-13-01412-f014:**
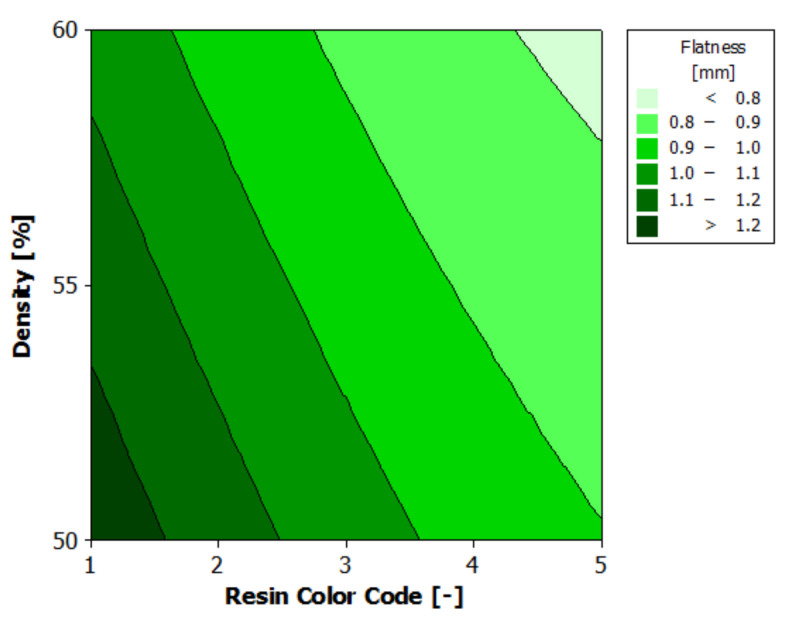
Contours of estimated response surface for flatness.

**Figure 15 polymers-13-01412-f015:**
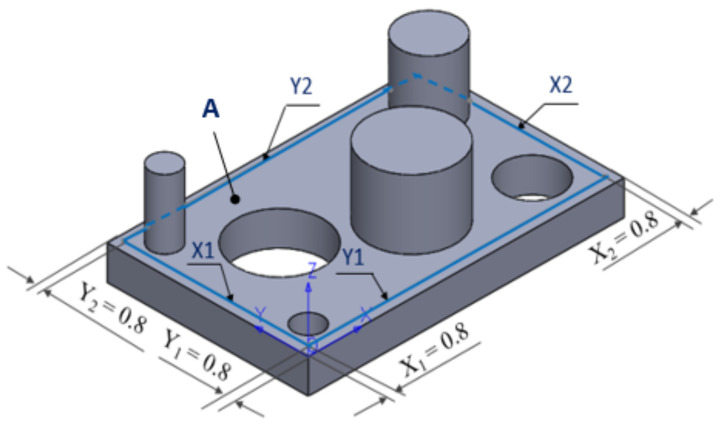
Definition of the measured linear contours.

**Figure 16 polymers-13-01412-f016:**
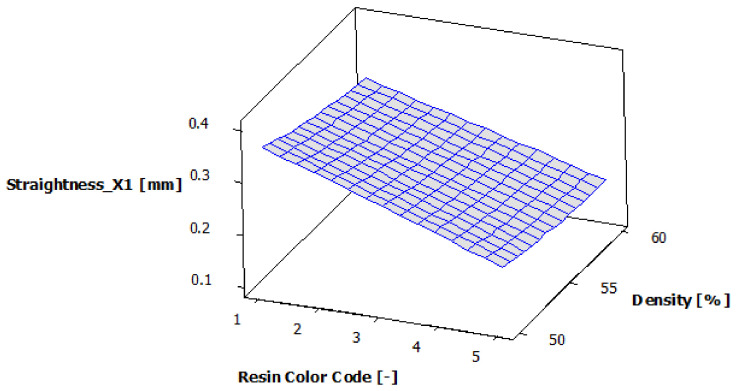
Estimated response surface for Straightness_X1.

**Figure 17 polymers-13-01412-f017:**
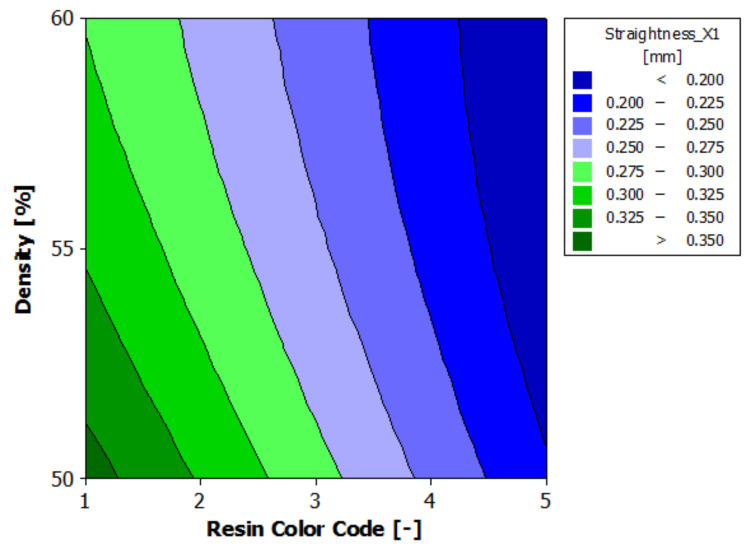
Contours of estimated response surface for Straightness_X1.

**Figure 18 polymers-13-01412-f018:**
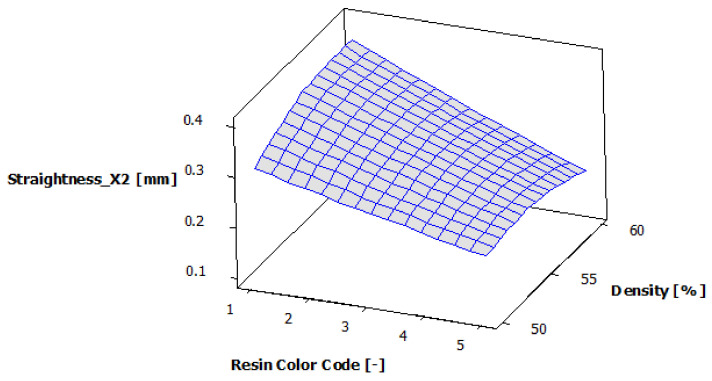
Estimated response surface for Straightness_X2.

**Figure 19 polymers-13-01412-f019:**
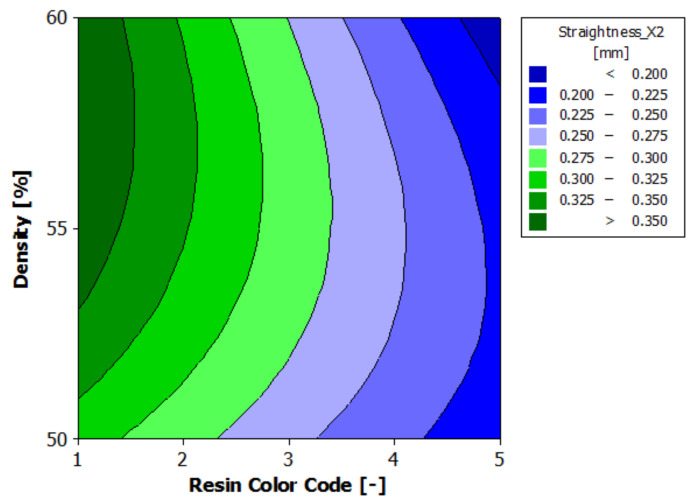
Contours of estimated response surface for Straightness_X2.

**Figure 20 polymers-13-01412-f020:**
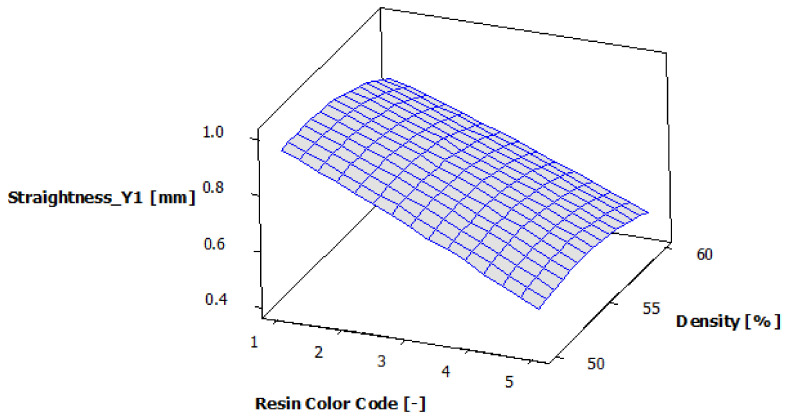
Estimated response surface for Straightness_Y1.

**Figure 21 polymers-13-01412-f021:**
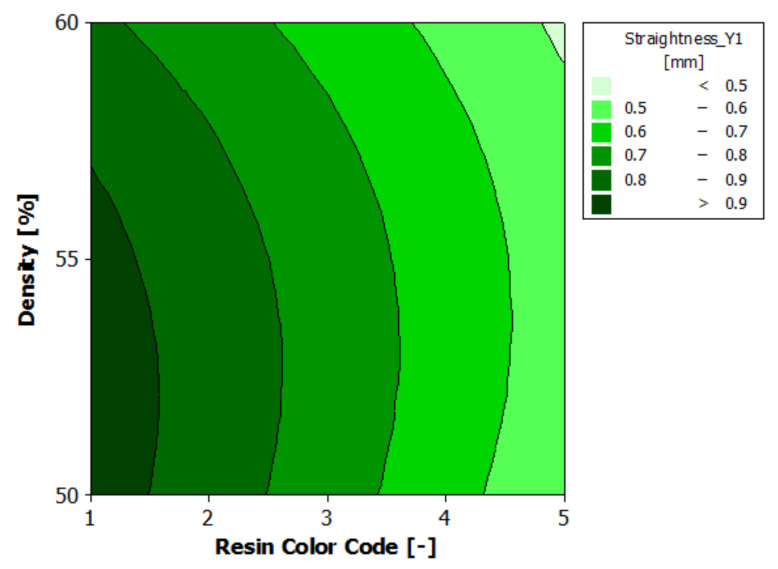
Contours of estimated response surface for Straightness_Y1.

**Figure 22 polymers-13-01412-f022:**
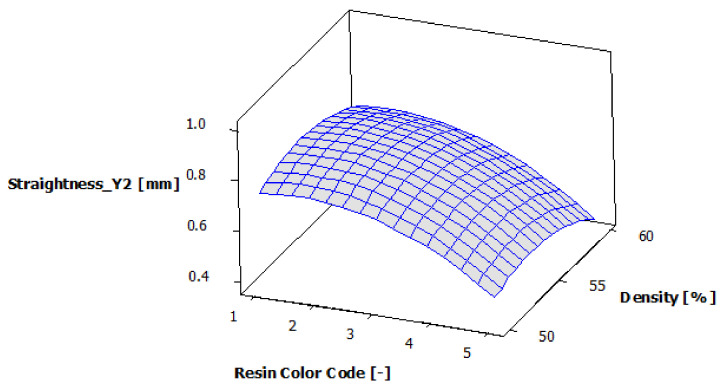
Estimated response surface for Straightness_Y2.

**Figure 23 polymers-13-01412-f023:**
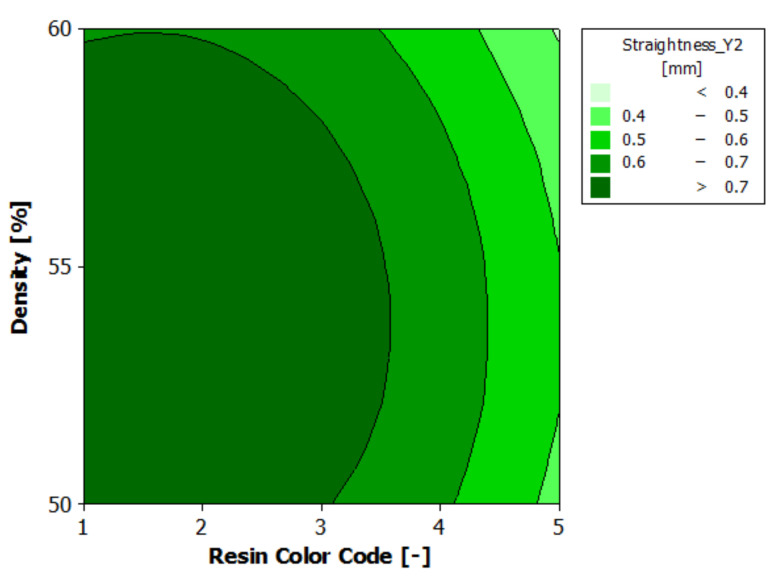
Contours of estimated response surface for Straightness_Y2.

**Figure 24 polymers-13-01412-f024:**
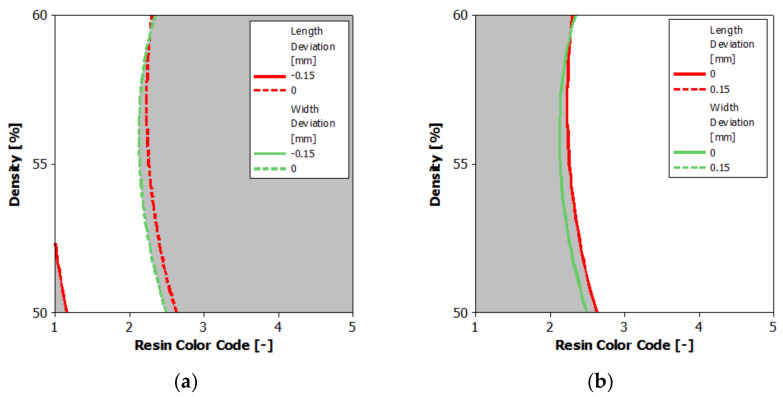
Overlaid contour plot of length deviation and width deviation: (**a**) negative deviations; (**b**) positive deviations.

**Figure 25 polymers-13-01412-f025:**
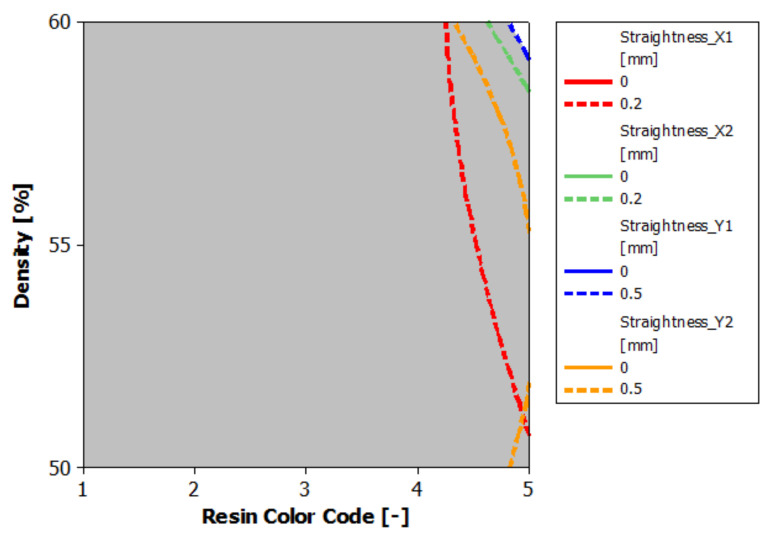
Overlaid contour plot of Straightness_X1; X2; Y1; Y2.

**Figure 26 polymers-13-01412-f026:**
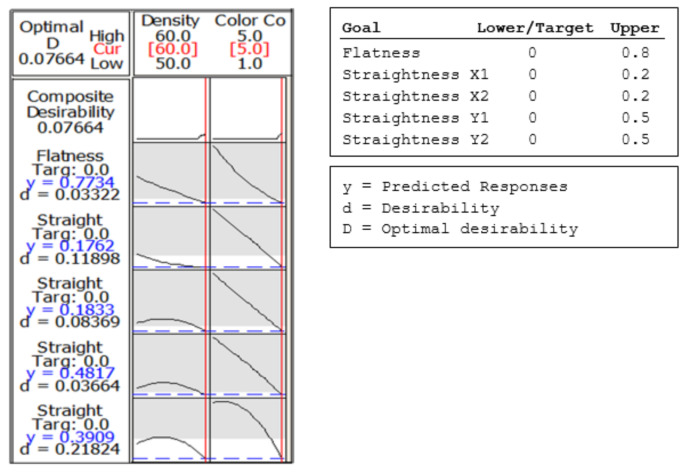
Optimization plot of Flatness and Straightness_X1; X2; Y1; Y2.

**Table 1 polymers-13-01412-t001:** The input data for the generation of the part.

Number	Type Top Support	Radius (mm)	Length (mm)	Contact Depth (mm)	Type Middle Support	Type Bottom	Radius (mm)	Density (%)	Layer Number
1	Conical	0.60	3.00	0.20	Cylinder	Skate	5.00	50	370
2	Conical	0.63	3.00	0.25	Cylinder	Skate	5.00	55	370
3	Conical	0.80	4.00	0.30	Cylinder	Skate	7.00	60	390

**Table 2 polymers-13-01412-t002:** Definition of the part probing strategy.

Physical Feature	Contact Depth (mm)	Z-Coordinate of the Measuring Plane (mm)	N° of Probing Points
Surface A	Plane	0	110
Surface B	Straight Line	−2	14
Surface C	Straight Line	−2	12
Surface B1	Straight Line	−2	14
Surface C1	Straight Line	−2	12
Contour X1	Straight Line	0	35
Contour X2	Straight Line	0	35
Contour Y1	Straight Line	0	55
Contour Y2	Straight Line	0	55

**Table 3 polymers-13-01412-t003:** Input data for the generation of the printed part and the size measuring results.

Number	Color	Color Code	Density (%)	Length (mm)	Width (mm)	Deviation (%)		Obs.
						Length	Width	
1.	2.		3.	4.	5.	6.	7.	8.
1	Green	1	50	54.786	34.821	−0.214	−0.179	
2		1	55	54.936	34.978	−0.064	−0.022	
3		1	60	54.823	3.873	−0.177	−0.127	
4	Blue	2	50	54.984	34.983	−0.016	−0.017	
5		2	55	54.985	35.007	−0.015	0.007	
6		2	60	55.016	34.995	0.016	−0.005	
7	Violet	3	50	54.977	35.007	−0.023	0.007	
8		3	55	54.994	35.003	−0.006	0.003	
9		3	60	54.992	34.977	−0.008	−0.023	
10	Black	4	50	55.111	35.113	0.111	0.113	
11		4	55	55.119	35.090	0.119	0.090	
12		4	60	55.125	35.091	0.125	0.091	
13	Clear	5	50	55.076	35.076	0.076	0.076	
14		5	55	55.033	35.010	0.033	0.010	
15		5	60	55.079	35.057	0.079	0.057	

**Table 4 polymers-13-01412-t004:** Analysis of variance for length deviation.

Source	Degrees of Freedom (Df)	Sum of Squares (SS)	Mean Square (MS)	F-Value	*p*-Value
Resin Color	4	0.123243	0.030811	19.65	0.000
Density	2	0.001928	0.000964	0.61	0.564
Error	8	0.012543	0.001568		
Total	14	0.137714	0.030811		

**Table 5 polymers-13-01412-t005:** Analysis of variance for width deviation.

Source	Degrees of Freedom (Df)	Sum of Squares (SS)	Mean Square (MS)	F-Value	*p*-Value
Resin Color	4	0.071184	0.017796	9.41	0.004
Density	2	0.001121	0.000561	0.30	0.751
Error	8	0.015137	0.001892		
Total	14	0.087442	0.017796		

**Table 6 polymers-13-01412-t006:** The output data for the flatness of the printed part.

Number	Resin Color	Color Code	Density (%)	Flatness (mm)	Obs.
1.	2.		3.	4.	8.
1	Green	1	50	1.182	
2		1	55	1.240	
3		1	60	1.035	
4	Blue	2	50	1.161	
5		2	55	1.044	
6		2	60	0.949	
7	Violet	3	50	1.217	
8		3	55	1.121	
9		3	60	0.911	
10	Black	4	50	0.763	
11		4	55	0.727	
12		4	60	0.722	
13	Clear	5	50	1.026	
14		5	55	0.776	
15		5	60	0.885	

**Table 7 polymers-13-01412-t007:** Analysis of variance for flatness.

Source	Degrees of Freedom (Df)	Sum of Squares (SS)	Mean Square (MS)	F-Value	*p*-Value
Resin Color	4	0.33395	0.083489	12.25	0.002
Density	2	0.07178	0.035891	5.27	0.035
Error	8	0.05451	0.006814		
Total	14	0.46024			

**Table 8 polymers-13-01412-t008:** The output data for the straightness of the printed part in the X and Y direction.

Number	Resin Color	Color Code	Density (%)	Straightness (mm)				Obs.
				X1	X2	Y1	Y2	
1.	2.		3.	4.	5.	6.	7.	8.
1	Green	1	50	0.357	0.318	0.971	0.664	
2		1	55	0.312	0.370	0.937	0.862	
3		1	60	0.318	0.378	0.798	0.666	
4	Blue	2	50	0.329	0.257	0.796	0.762	
5		2	55	0.270	0.303	0.795	0.749	
6		2	60	0.246	0.307	0.780	0.723	
7	Violet	3	50	0.308	0.286	0.825	0.805	
8		3	55	0.282	0.319	0.825	0.829	
9		3	60	0.258	0.294	0.669	0.595	
10	Black	4	50	0.223	0.237	0.575	0.568	
11		4	55	0.193	0.224	0.576	0.528	
12		4	60	0.195	0.212	0.570	0.530	
13	Clear	5	50	0.198	0.193	0.533	0.480	
14		5	55	0.209	0.239	0.603	0.509	
15		5	60	0.174	0.186	0.464	0.443	

**Table 9 polymers-13-01412-t009:** Analysis of variance for straightness_X1.

Source	Degrees of Freedom (Df)	Sum of Squares (SS)	Mean Square (MS)	F-Value	*p*-Value
Resin Color	4	0.039901	0.009975	38.06	0.000
Density	2	0.005200	0.002600	9.92	0.007
Error	8	0.002097	0.000262		
Total	14	0.047198			

**Table 10 polymers-13-01412-t010:** Analysis of variance for straightness_X2.

Source	Degrees of Freedom (Df)	Sum of Squares (SS)	Mean Square (MS)	F-Value	*p*-Value
Resin Color	4	0.043758	0.010939	24.73	0.000
Density	2	0.002692	0.001346	3.04	0.1047
Error	8	0.003538	0.000442		
Total	14	0.049988			

**Table 11 polymers-13-01412-t011:** Analysis of variance for straightness_Y1.

Source	Degrees of Freedom (Df)	Sum of Squares (SS)	Mean Square (MS)	F-Value	*p*-Value
Resin Color	4	0.29095	0.072738	33.68	0.000
Density	2	0.02559	0.012796	5.93	0.026
Error	8	0.01728	0.002159		
Total	14	0.33382			

**Table 12 polymers-13-01412-t012:** Analysis of variance for straightness_Y2.

Source	Degrees of Freedom (Df)	Sum of Squares (SS)	Mean Square (MS)	*F*-Value	*p*-Value
Resin Color	4	0.19670	0.049174	11.09	0.002
Density	2	0.02755	0.013776	3.11	0.100
Error	8	0.03546	0.004432		
Total	14	0.25971			

## Data Availability

The data presented in this study are available on request from the corresponding author.

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
