# Peer review of "Comparative Study of the Influence of Bio-Resin Color on the Dimension, Flatness and Straightness of the Part in the 3D Printing Process"

_polymers, 2021, doi:10.3390/polym13091412_

Round 1

Reviewer 1 Report

The paper "Comparative Study of the Influence of Bio-Resin Color on the 2 Dimension, Flatness and Straightness of the Part in the 3D 3 Printing Process" reports the study of printing parameters to obtain good resolution pieces with different colors.

The paper results more a technical report than a scientific study and I found it quite hard to follow both because some points need to be better explained and because the reader needs to have technical knowledge of standards.  In my opinion this kind of work would be more suitable for an applied and technical journal. 

Furthermore I have some observations:

The authors name the bio-origin of the resins in the title but then this is not discussed in the paper, commercial materials are used. Some indications about the resin nature should be done. Are the author using always the same resin with different added dyes or are these different products with different compositions?

In the experimental plan (line 79) the authors define the best exposure time but then a further study is reported. What are the first exposure times referred to?

The layer thickness is also another fundamental parameter that can influence the polymerization and thus the resolution, above all when different dyes (that could lead to different light penetration/scattering) are considered. This should be also considered.

Then english should be improved and the introduction enriched.

In my personal opinion this paper is not suitable for polymers 

Reviewer 2 Report

this paper need revision before publish

  1. more chemical structure about the resin and color should provide, because it will affect all the process and also will relate with the results, and it is important to explain the results.
  2. the figure 1,2 and 3 are not necessary, it did not provide important information.
  3.   figure 9, 14, 17 could be put in figure 5
  4.  the explain of the results is not very clear, it did not look inside the polymerization process and the materials properties
  5. the paper looks more like a experimental report, it should write more scientific
  6. the references should keep the same format

Reviewer 3 Report

There are several issues in the manuscript that should be addressed before further consideration for publication.

  1. Suggest to use ISO/ASTM standard terminology for the processes. There should be consistency in naming the processes within the manuscript.
  2. As the focus is on the color resins, there should be clear description of the difference in them, for example, their compositions etc
  3. How are the features of the benchmark part decided? Any consideration on the other features for consideration such as curvature?
  4. There has been numerous work done on benchmarking for 3D printing. Any references from them? What is the new insights from this?
  5. Are the process parameters more significant than the color of the resins in affecting the results?

Reviewer 4 Report

Dear Editor and Authors,

The manuscript addresses the problems of 3D printing. The authors want to address the problem of colour as a determining factor in 3D printing. From Figure 1 you can learn that these are resins that are generally available. The fundamental question, which is the main curiosity of the reviewer, is. What ingredients besides those that impart colour do these resins have in addition. Are they the same or do they contain something extra? The reviewer cannot comment on the whole study without this data. Therefore, I ask you to complete this data and submit the manuscript for a second review.

Round 2

Reviewer 1 Report

I still find the paper very technical but I understand that it is in line with other published papers.

I appreciated the improvements done, thus, if the editor finds the manuscript suitable for the journal, I agree with its publication.

Reviewer 2 Report

Accept

Reviewer 3 Report

NA

Reviewer 4 Report

Dear Authors,

Thank you for your revision. After the second evaluation I would like to ask the Editor to accept your contribution for publication.